# Mitigation of Flood Risks with the Aid of the Critical Points Method

Karel Drbal [1], Miroslav Dumbrovský [2], Zlatica Muchová [3,*], Veronika Sobotková [2], Pavla Štěpánková [1] and Bořivoj Šarapatka [4]

1   T. G. Masaryk Water Research Institute, p.r.i., 60200 Brno, Czech Republic; karel.drbal@vuv.cz (K.D.); pavla.stepankova@vuv.cz (P.Š.)
2   Institute of Water Landscape Management, Brno University of Technology, 60200 Brno, Czech Republic; dumbrovsky.m@fce.vutbr.cz (M.D.); sobotkova.v@fce.vutbr.cz (V.S.)
3   Institute of Landscape Engineering, Slovak University of Agriculture in Nitra, 94901 Nitra, Slovakia
4   Department of Ecology and Environmental Sciences, Palacký University Olomouc, 77146 Olomouc, Czech Republic; borivoj.sarapatka@upol.cz
*   Correspondence: zlatica.muchova@uniag.sk

**Abstract:** Concentrated surface run-off caused by torrential rain leads to the transport of sediments from soil erosion processes within catchment areas of critical points, which represents a basic component of flood risks. Clear identification of such critical points offers a basis for a suitable new strategy of threat mitigation, via both organizational and structural measures in catchment areas. Critical points are determined in places where generated paths of concentrated surface run-off cross given boundaries of built-up areas. The threshold values for the catchment area of a critical point were derived on the basis of hydrological calculations, field research, and the evaluation of hundreds of specific manifestations of damage in built-up areas for particular critical profiles. The characteristics were determined as follows: contributing area between 0.3 and 10 km$^2$, average slope more than 3.5%, and percentage of arable land more than 40%. Indicator F was determined for the distribution of the territory of the Czech Republic according to the risk of flooding. Knowledge of the existence of critical points enables the implementation of preventive measures, the evaluation of flood risk associated with the transport of sediment due to erosion processes, and the improvement of management measures in respective catchment areas, even before an event occurs. The proposed procedure outputs shall be reflected in spatial planning documentation, land consolidation, and catchment area management plans. Incorporation of critical points into open-access public web-maps can help with assessing the semi-quantitative expression of risk to built-up areas arising from the threat of local flooding.

**Keywords:** concentrated surface run-off; mitigation of flood risks; transport of sediments; soil degradation; water erosion; water retention



## 1. Introduction

The transport of sediments as a product of soil erosion processes within catchment areas is a significant issue for agricultural land in particular. The floods that affected large areas of the Czech Republic (CZ) in recent decades have raised many questions and contradictory views on further progress in the field of water management within the landscape, in particular, the issue of river basin protection against the adverse effects of flash flooding during extreme hydrological events. Direct losses due to flooding [1] take the form of damage to buildings, industrial areas [2], and urban and landscape infrastructure such as roads and railways, including damage to arable land due to water erosion on sloping areas [3] and land in the path of concentrated surface run-off [4,5]. These types of loss are evaluated using financial terms, if possible, or by other forms of classification [6,7].

The importance of the topic in CZ is highlighted by the consequences of the catastrophic flooding in July 2009, which affected most regions of northern and eastern Moravia, causing great material damage [8,9]. Directive 2007/60/EC (2007) frames the evaluation and reduction of flooding impact. Infrastructural measures [10] that protect the landscape from torrential water and waterlogging and provide sources of water to cover the moisture deficit are reservoirs, polders, drainage and irrigation, etc. They are often combined with structures to protect the soil from wind and water erosion, such as grassing, afforestation, windbreaks, infiltration belts, terraces, dams, and partitions. Their implementation, due to the fragmentation of land ownership, is possible especially after the implementation of land consolidation and can be financed from resources derived from the state or municipalities and cities.

Targeted implementation of preemptive measures based on the evaluation of flood risk associated with the transport of sediments due to erosion processes, and the subsequent improvement of agricultural procedures in respective catchment areas, can prevent substantial damage even when a torrential rain event occurs. The higher frequency of extreme rain leads to increased risk of flooding and the accompanying problems, with sediment floating from intensively managed land units to urbanized areas of towns and villages. Some of the currently prevailing land use trends tend to worsen the situation.

It is expected that, in CZ, extreme climatic phenomena will become more frequent, with unusually dry periods, due to increased air temperature and a lack of precipitation, as well as intense downpours affecting the landscape and causing environmental damage [11]. A different approach [12–14] is needed to cope with events relating to changing climatic conditions.

Climate change at the European level is addressed in the study "Climate change: impacts and vulnerabilities in Europe" [15]. This document assesses the development of the climate in Europe up to the year 2100 in detail. Increasing frequency of heavy torrential rains is associated with a higher frequency of local flash floods. Flood damage is on the rise in Europe. The risk of intense water erosion is also connected to extreme precipitation and floods. According to a representative set of regional climate models (EURO-CORDEX, https://www.euro-cordex.net, accessed on 11 January 2022), the intensity of observed manifestations of ongoing climate change in Europe, including the Czech Republic, will increase in the coming decades. Erosion and runoff conditions are influenced, in addition to the increase in temperature, by the changing characteristics of precipitation and hydrological balance of the river basins. The authors of [16,17] show, e.g., that the average precipitation total of an erosion event remains largely constant, while the duration of the precipitation event shortens with increasing intensity and erosion efficiency.

Due to the risk of floods and increasing damage, the European Commission introduced the Floods Directive, EC 2007 [18], on the assessment and management of flood risks. European flood risk regulation adopts a bioregional scale and concentrates on the river basin level in particular [19]. With the growing risk of floods due to climate change and socio-economic trends, governments are under constant pressure to continue to implement flood protection measures, and the impact of extreme floods on individual regions is increasing significantly [20]. Flood risk mapping is important for flood risk mitigation planning, which can be both international [21–23] and national, e.g., on the territory of the Czech Republic or in a sub-basin [8,24]. In the landscape, however, it is not only floods, but also soil erosion and its further damage that are of concern; these can be expressed, e.g., by soil risk maps [25].

Even areas with no flowing water or bodies of water are endangered by flooding, as witnessed in recent decades in CZ, where sudden downpours caused erosion and flooding. Such flooding can bring extensive material damage and even loss of human life in the affected areas and devastation of the cultural environment. More than one hundred locations during the flood in 2009 were identified as being in the Jičínka and Luha watershed, where built-up areas were affected by surface run-off [26–28]. Similar studies have not been performed elsewhere in CZ; only the identification of ephemeral gullies within an area of agricultural land was carried out for limited number of watersheds [29]. Grešková [30]

describes the factors influencing flood risk for small-sized basins. A geographical approach to assessing flood risk is formulated by Zeleňáková [31]. However, the assessment of classification variables for sustainable flood retention measures is discussed in many sources. The authors of [32,33] mention as many as 40 anticipated variables that can be localized. Gunnell et al. [34] evaluated natural infrastructure for flood management, introducing metrics for flood buffering. The authors of [35] constructed such criteria as drainage system condition, distance from river, topographic condition, land use, road network condition, etc. The fundamental problem in multi-criteria evaluation is the credibility of weighting for parameters [36–39]. Roub et al. [40] propose a model for catchment, considering criteria relating to land cover.

Flash flooding due to torrential rainfall occurs mainly in small watersheds and is quite frequent in CZ [41]. Flash flooding has a site-specific character due to causal factors and the physical-geographical parameters of basins [42,43]. Correct identification of critical points and characteristics of catchment areas, including the estimation of potential damage, is an important basis for the subsequent design of conservation measures to reduce the risks of adverse consequences of concentrated surface run-off [44]. A related topic was studied in Slovakia [42].

Many of the preemption and mitigation concepts are based on an integrated approach [45–49]. Green measures for the natural retention of flood waters are gaining importance as an aspect of comprehensive efforts in the landscape with public support [50].

The Geographical Information System (GIS) may provide important information by determining areas vulnerable to flooding based on combining data and simultaneous analysis of many variables [51–54].

This article's main aim is assessing the risk of flooding due to torrential rain by identifying and evaluating critical points (CPs) and areas forming concentrated surface run-off. The presented approach evaluates the potential contributing area of an event in relation to the most vulnerable localities, i.e., urbanized land. Among the characteristics of the contributing area are the physical-geographical parameters of catchments, pedological and hydrological descriptors, and land use. Application of the CP method leads to the identification of the zones (areas, hydrological units, etc.) that are threatened, to a greater or lesser extent, by flash flooding.

## 2. Materials and Methods

### 2.1. Study Area

The Czech Republic is complex in its geological structure; highly variable in relief, it represents the practical roof of Europe in hydrographic terms and is a significant spring source for the European continent. CZ (Figure 1) is divided into three main European watersheds (watersheds of main watercourses, i.e., first order watersheds): The Elbe watershed (denoted 1) drains water into the North Sea, the Odra watershed (denoted 2) drains water into the Baltic Sea, and the Danube watershed (denoted 4) drains water into the Black Sea.

Rivers (with their basins) of the first order run into the sea; rivers (with their basins) of the second order are their direct tributaries; rivers (with their watersheds) of the third order are tributaries of rivers of the second order; streams, brooks, creeks (with their catchments) of the fourth order are tributaries of rivers of the third order. Sub-watersheds of main watercourses are marked as watersheds of the second order (defined 114, 204, 415, etc.). Watersheds of the third order are regarded as the basic watersheds, and the last group of watersheds are those of the fourth order. The study used the designation of the second order basin, i.e., the first three digits of the eight-digit code (0-00-00-000). Calculations were also organized by so-called river basin groups, for example, 201–202–203. The first place determines the appurtenance to the watershed of the main river of the first order. The second two places define the appurtenance to the partial sub-watershed of the main river.

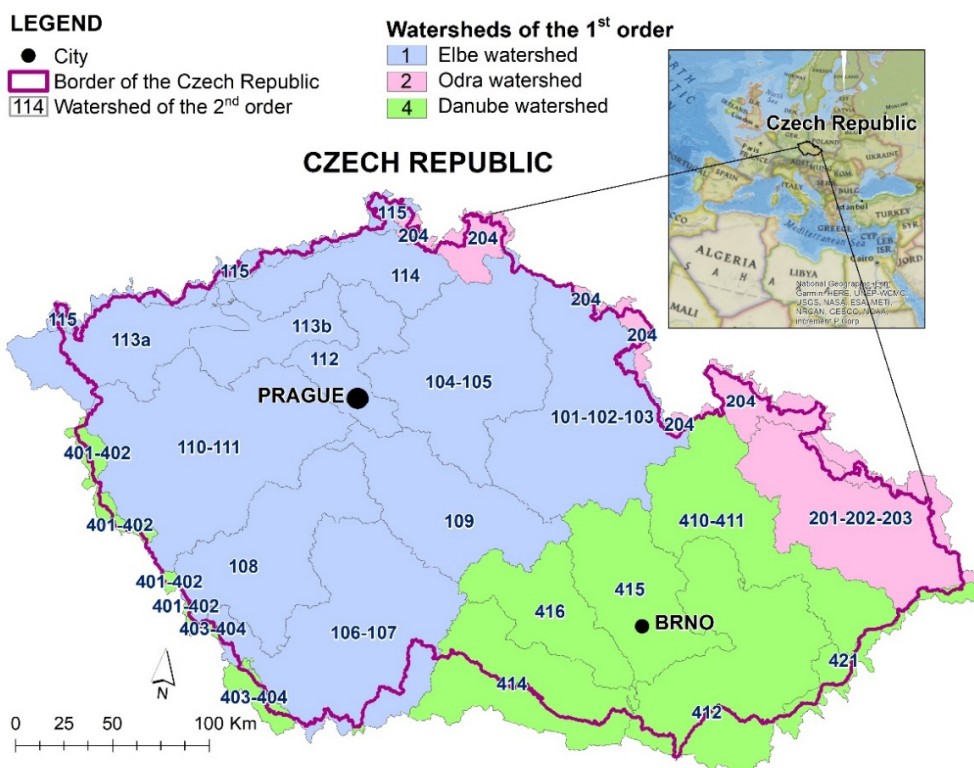

**Figure 1.** The study area and an overview of watersheds and groups of watersheds of the second order.

The basis for assessing the risk of floods due to torrential rains is the identification and evaluation of critical points (CPs) and areas where concentrated surface runoff is formed. The most vulnerable localities, the urbanized areas, require focusing on the main source of danger, i.e., the contributing (source) areas of critical points with relevant characteristics as physical-geographical parameters of the basin, pedological and hydrological descriptors, and land use. The key is the application of the CP method to identify zones (areas, hydrological units, etc.) that are threatened by flash floods. This identification of larger hydrological units with a specified degree of endangerment is important for prioritizing further steps that lead to the management of flash flood risks in urbanized localities.

The subject of the study was the entire territory of the Czech Republic. The CP catchment areas and evaluation of relief, especially slope conditions, were generated on the basis of a digital relief model of the 4th generation (DMR 4G, in the altitude reference system Balt after adjustment with a mean height error of 0.3 m in exposed and 1 m in wooded terrain). Data on precipitation were obtained from the national database of the Czech Hydrometeorological Institute (CHMI, https://www.chmi.cz, accessed on 21 January 2022), information on the physical properties of the soil were derived from rated soil-ecological units (BPEJ, https://bpej.vumop.cz/, accessed on 21 January 2022), the landscape cover was determined according to the public land register (https://eagri.cz/public/app/lpisext/lpis/verejny2/plpis/, accessed on 21 January 2022), and orthophotomaps were verified by field surveys.

### 2.2. Determination of Critical Points

Critical points, CPs, are determined [54] in places where generated paths of concentrated surface run-off (PCR) enter urbanized spaces (B) (Figure 2). A CP is given by an intersection of the given boundary of a built-up area with paths of concentrated surface run-off whose catchment area (*Ac*) is between 0.3 and 10 km$^2$ [27,28].

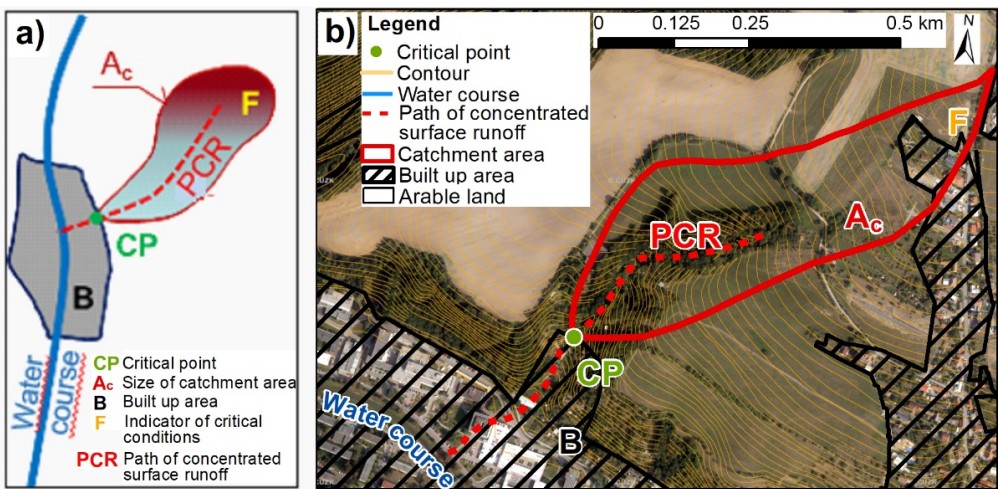

**Figure 2.** Identification of a critical point at the intersection of a path of concentrated surface run-off and a built-up area boundary, and the size of catchment area with the indicator of critical conditions: (**a**) diagram and (**b**) orthophotomap.

The principle of determining critical points is outlined in Figure 2. A critical point is determined by an intersection of the given boundary of a built-up area with paths of concentrated surface run-off. Flooding risk arises from the threat to a built-up area lying below a given critical point. Built-up areas include whole urban zones (especially residential buildings).

The threshold values (size of the contributing area, slope conditions, and percentage of arable land) were determined on the basis of field surveys and the evaluation of hundreds of specific manifestations of damage in the built-up areas at particular critical profiles. A maximum area of 10 km$^2$ was set on the basis of the limitation of the Curve Number (CN) method used.

Combined criteria I are recommended for identification of CP:

1. C1—size of catchment area (*Ac*) ($0.3 \leq Ac \leq 10.0$ km$^2$)
2. C2—average slope (*Ip*) of catchment area ($Ip \geq 3.5\%$)
3. C3—percentage of arable land (*AL*) in catchment area ($AL \geq 40\%$)

Combinations of physical-geographical situation, variation in land cover, land use, and potential for extreme precipitation in specific catchment areas are expressed by an indicator of critical conditions for flooding, *F* (Formula (1)).

$$F = A_{c,r} \cdot H_{m,r} \cdot (a_1 \cdot I_p + a_2 \cdot AL + a_3 \cdot CNII) \tag{1}$$

where *F* is the indicator of critical conditions (dimensionless), $a_i$ ($I$ = 1, 2, 3) are the model parameters ($a_1 = 1.48876$; $a_2 = 3.09204$; $a_3 = 0.467171$), $A_{c,r}$ is the relative size of the catchment (normalized to the maximum of 10 km$^2$) (dimensionless), $I_p$ is the average slope (%), *AL* is the percentage of arable land (%), *CNII* is CN values (AMCII—average level of saturation of soil) for the Czech Republic, $H_{m,r}$ is the relative sum of one-day precipitation with a return period of 100 years (dimensionless).

Limiting value of F forms the fourth criterion condition:

4. C4—indicator of critical conditions ($F \geq 1.85$)

The value of 1.85 resulted from the evaluation of an extensive set of CP basin characteristics from the pilot areas of the Luha, Jičínka, and Husí potok watercourses that were affected by the floods in 2009. The principles of the critical points method were tested on these data.

The Czech Hydrometeorological Institute is the provider of data for the determination of CNII and $H_{m,r}$. Research carried out on model floods indicated damage in catchments with percentage of arable land less than 40% or entirely wooded area; thus, the selection

made according to criteria conditions C1 to C4 was extended to CPs with catchment area of 1 km$^2$ and above, together with an average slope of 5% and above:

- C1A size of catchment area (*Ac*) ($1.0 \leq Ac \leq 10.0$ km$^2$)
- C2A average slope (*Ip*) of catchment area ($Ip \geq 5\%$)

Output from the CP identification process is a graph showing the locations of the identified CPs and their catchment areas, including the selected characteristics (size and average slope of catchment area, percentage of arable land in a catchment area, etc.).

*2.3. Evaluation of Influence of Critical Conditions within Catchment Areas on Run-Off Characteristics in Watersheds*

The level of influence of critical point watersheds within watersheds of higher hydrological orders was verified by means of Least Trimmed Squares robust regression (LTS regression).

Using this statistical method, three basic robust models, founded on the aforementioned data, were proposed to estimate the weighted average of the indicator of critical conditions for hydrological units with available CN values.

The evaluation looked at the dependence of the level of urbanization on the size of watersheds, finding a general relationship for the calculation of new characteristics, which would determine the value of indicators of critical conditions specifically for watersheds of a higher order (Formula (2)).

$$B = a \cdot A_w \tag{2}$$

where *B* is the size of built-up areas with residential buildings ($10^{-2}$ km$^2$), $A_w$ is the size of the watershed (km$^2$), and a is the model parameter ($a = 0.536$) determined by the LTS regression method.

A precision of 87% (correlation of determination R$^2$ = 0.87) shows the direct proportion of urbanization level of a territory by residential objects, depending on the size of the catchment area. It was extremely important to verify this relationship because the basic principle of the methodology of CP identification relates to the determination of built-up areas within catchment areas, a characteristic that objectively tends to be independent of hydrological conditions rather than physical-geographical conditions.

The second model was used to test the dependence of values of a specific flow rate from catchment areas from the respective return period generated from the catchment areas, which belong to CPs within a hydrological unit, on selected explanatory variables (Formula (3)).

$$q_{100r} = a_1 \cdot A_w + a_2 \cdot q_{100} + a_3 \cdot B \tag{3}$$

where $q_{100r}$ is the reduced value (by the ratio of catchment areas of CP and the size of watershed) of a specific flowrate with a return period of 100 years (m$^3 \cdot$s$^{-1} \cdot$km$^{-2}$), *B* is the size of built-up areas with residential buildings ($10^{-2}$ km$^2$), $A_w$ is the size of the watershed (km$^2$), and $a_i$ (i = 1, 2, 3) are the model parameters ($a_1 = -0.002$; $a_2 = 0.36$; $a_3 = 0.0037$).

There was a precision of 79% (multiple correlation of determination R$^2$ = 0.79) of occurrence of a random event, which is the result of the influence of drainage ratios in closing profiles of hydrological units up to 150 km$^2$ only in relation to the ratio of catchment areas of CPs to the size of a watershed.

The third model was used to verify the closeness of relations between the weighted average of the indicator of critical conditions ($F_r$) using the reduced ratio of catchment areas of CPs and the size of the watershed and reduced value of a specific flow rate with a return period of 100 years ($q_{100r}$) and catchment area urbanization (*B*), i.e., total area of residential buildings (Formula (4)).

$$F_r = a_1 \cdot q_{100r} + a_2 \cdot B \tag{4}$$

where *Fr* is the reduced value of the weighted average of the indicator of critical conditions (dimensionless variable), $q_{100r}$ is the reduced value (using the ratio of catchment areas of CP and the size of watershed) of a specific flowrate with a return period of 100 years

($m^3 \cdot s^{-1} \cdot km^{-2}$), $B$ is the size of built-up areas with residential buildings ($10^{-2}$ $km^2$), and $a_i$ (i = 1, 2) are the model parameters ($a_1$ = 7.193; $a_2$ = 0.0644).

Almost 73.5% (multiple correlation of determination $R^2$ = 0.735) of the occurrences of a random event can be successfully explained by Formula (4).

The robust models confirmed relevance, in particular the determination of a reduced critical condition indicator value. Nevertheless, the ambiguities that arise in the approximations of relations by the stated models (even though robust models were used) do not justify the use of the models (2 to 4) for analogical application on other data and in other catchment areas. The next step was the calculation of reduced values of the weighted average of the indicator of critical conditions and other explanatory variables using the procedure for the statistical evaluation as in models 2 to 4.

## 3. Results and Discussion

The CP identification process enabled us to determine 9261 CPs. An overview of territorial representation of the generated CPs in the Czech Republic is shown in Table 1 (Figure 3). Subsequently, selections were made according to the known characteristics, i.e., the indicator of critical conditions ($F$) and the size of catchment areas ($Ac$). The aforementioned data contain the basic information for the visualization of the degree of the potential impact of flood hazards from torrential rain and also facilitation of the main goal, which is a semi-quantitative expression of the extent of risk to urbanized spaces arising from local flooding hazards.

**Table 1.** Overview of territorial representation of generated CPs in the Czech Republic.

| Region | Number of CPs | Catchment Area of CPs ($km^2$) |
| --- | --- | --- |
| South Bohemian Region | 804 | 1822 |
| South Moravian Region | 713 | 1507 |
| Karlovarský Region | 204 | 671 |
| Region Vysočina | 854 | 1852 |
| Královéhradecký Region | 708 | 1589 |
| Liberecký Region | 476 | 1209 |
| Moravian and Silesian Region | 727 | 1609 |
| Olomoucký Region | 683 | 1564 |
| Pardubický Region | 715 | 1563 |
| Plzeňský Region | 752 | 1671 |
| Prague | 24 | 59 |
| Central Bohemian Region | 1288 | 2776 |
| Ústecký Region | 669 | 1770 |
| Zlínský Region | 644 | 1688 |
| Total | 9261 | 21,350 |

The relations of the number of CPs to both mentioned variables, i.e., the indicator of critical conditions $F$ and the size of catchment area $Ac$, are shown in Figures 4 and 5, which illustrate the overall progress of the aforementioned dependences for the data for the whole of the Czech Republic. Figure 4 gives information about the number of CPs and the sum of their catchment areas. Figure 5 gives information about the number of CPs and the sum total of arable land within their catchment areas. This information is useful for local decision making, as it indicates the extent of the problem. For example, in an area with an F indicator lower than 15, it is not necessary to propose technical measures; "best management practices", such soil conservation agrotechnology, grassed waterways, or protective grassing/afforestation, are sufficient to reduce the peak flow. The threshold value was set on the basis of hydrological calculations performed on hundreds of CP contributing areas.

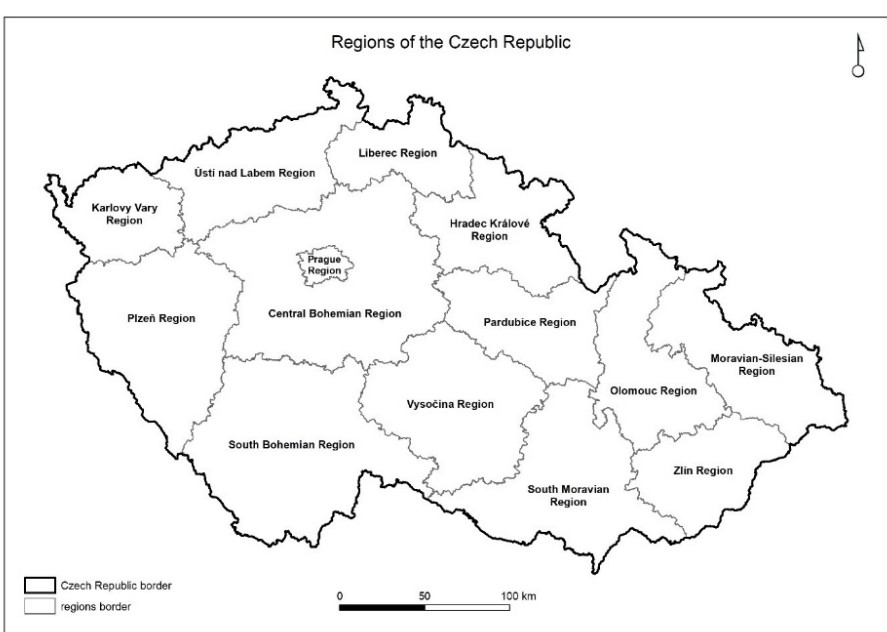

**Figure 3.** Regions of the Czech Republic.

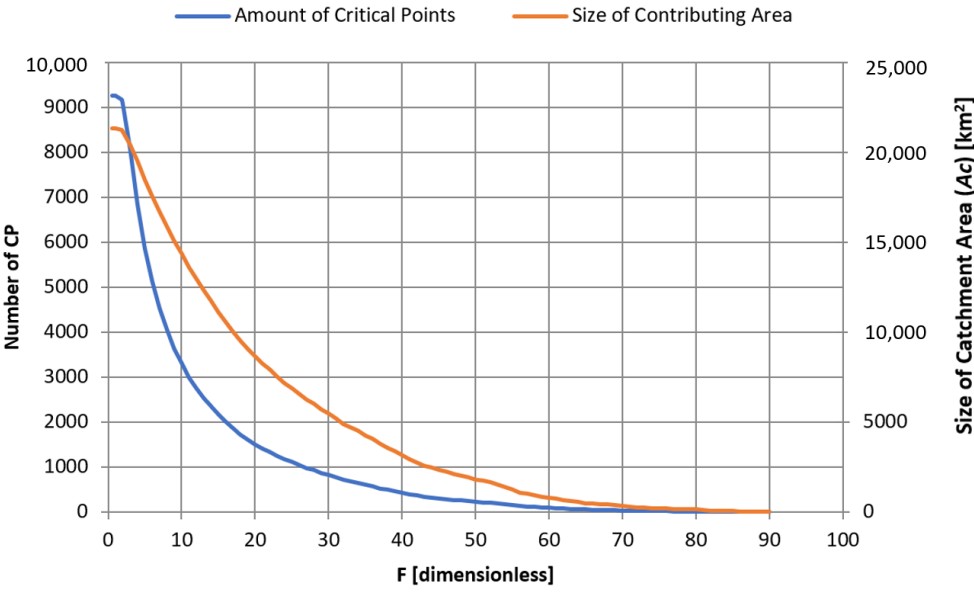

**Figure 4.** Relations of the number of CPs (blue curve) to the values of the indicator of critical conditions (*F*) and size of catchment areas (*Ac*).

The graph of dependences of the number of CPs on the indicator of critical conditions *F* and the size of arable land in a catchment area *AL* is given in Figure 5.

The analysis of soils and localities endangered by torrential rain, or more precisely by flooding caused by torrential rain, shows that the overall size of these endangered areas is relatively high, and for the whole of the Czech Republic it approximates to thousands of square kilometers.

Table 2 illustrates examples of selections of discrete values of the indicator of critical conditions (*F*) corresponding to the number of CPs, the total size of the catchment areas of CPs (*Ac*), and the area of arable land within catchment areas (*AL*).

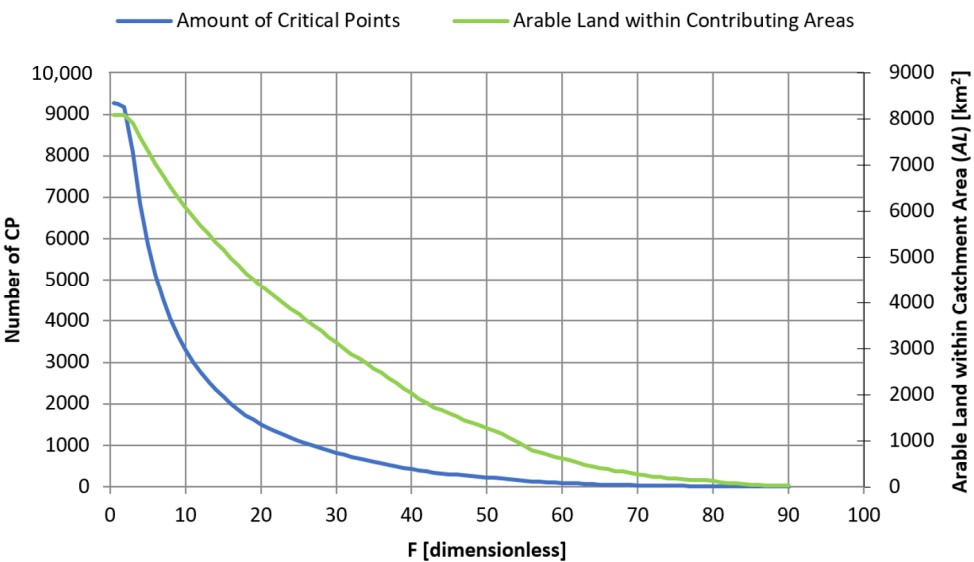

**Figure 5.** Graph of dependences of the number of CPs (blue curve) on the values of the indicator of critical conditions (*F*) and size of arable land within the catchment areas (*AL*).

**Table 2.** Examples of selection of CPs according to the value of the indicator of critical conditions *F*.

| *F* | Number of CP | Size of Catchment Areas (*Ac*) (km²) | Arable Land within Catchment Area (*AL*) (km²) |
|---|---|---|---|
| 37 | 526 | 3552 | 2217 |
| 40 | 427 | 2972 | 1914 |
| 45 | 305 | 2233 | 1519 |
| 52 | 206 | 1608 | 1123 |
| 59 | 101 | 836 | 631 |

Using a weighted indicator of critical conditions applied to a watershed of the fourth order, a regionalization of the level of risk to the territory by flooding from torrential rain was carried out, again for whole country. Almost 35% of the state (approx. 27,500 km²) lies outside the area of significant risk of this kind, and a further 40% of the territory corresponds to only a low level of risk from floods caused by torrential rains (Table 3). Less than a quarter of the territory of the Czech Republic falls within the categories of medium (18.3%) and high (5.7%) level of risk.

**Table 3.** The extent of territory with individual degree of risk of flash flooding.

| Risk Rate | Number of Watersheds of 4th Order | Average Size of Watersheds of 4th Order (km²) | Total Area (km²) | Share of Area (%) |
|---|---|---|---|---|
| High | 616 | 7.34 | 4524 | 5.7 |
| Middle | 1448 | 9.97 | 14,442 | 18.3 |
| Low | 1992 | 16.26 | 32,381 | 41.1 |
| Without any risk | 4899 | 5.62 | 27,520 | 34.9 |

The risk rate of flooding by torrential rain was determined on the basis of categorization of fourth order watersheds by means of reduced values of indicators of critical conditions. The results of the theoretical calculation were verified on pilot watersheds of the Luha, Jičínka and Husí potok [27,28]. These watersheds were severely affected by flooding by torrential rain in 2009, resulting in the loss of human life and extensive material damage.

Calculation of the reduced values of the indicator of critical conditions for watersheds of between 10–150 km² enabled the preparation of data files for watersheds or groups of

watersheds of the second order. In water basins with an area of 10–150 km$^2$, there is (still) a positive influence of the soil and water protection measures proposed and implemented in catchment areas of CPs. This represents (the positive influence of) the synergic effect of all proposed measures within a catchment area of a CP, in the context of river basins of 10–150 km$^2$.

The achieved results of relationships between values of the reduced indicator of critical conditions depending on size of watershed are shown in Figure 6 for the corresponding group of watersheds 201–202–203.

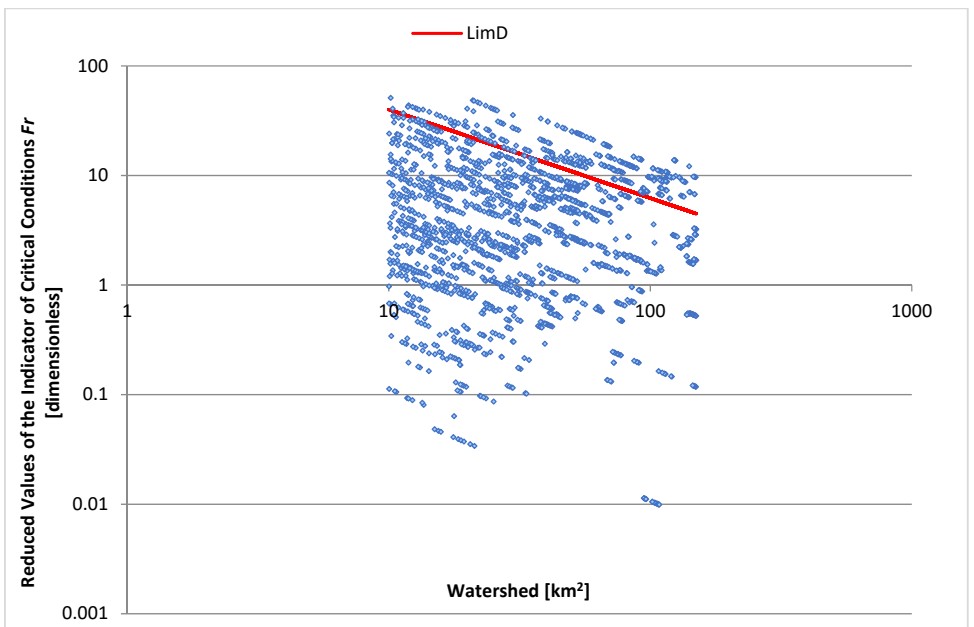

**Figure 6.** Reduced values of the indicator of critical conditions $F_r$ depending on the size of the watershed within the group of watersheds 201–202–203.

Conservation measures implemented in CP catchment areas will also contribute to positive changes in run-off rate in higher order watersheds. An area of 10 km$^2$ is the upper limit of the CP catchment area. In watersheds of over 150 km$^2$, the influence of flood prevention measures is insignificant or non-existent. Therefore, the determination of $F_r$ (reduced values of the indicator of critical conditions) was carried out in watersheds of between 10 and 150 km$^2$ throughout the Czech Republic.

The straight red line represents a proposed (testing) limit by which the primary selection of localities (catchment areas), with potentially the most destructive impact from flooding by torrential rain, was performed. These are watersheds that are suitable for considered proposals of even structural (costly) measures. The selection details are shown in the in Figure 7.

Figure 5 represents the relationship between hydrological units (watersheds) and reduced values of the indicator of critical conditions in a logarithmic scale. The red line—LimD (Figures 6 and 7), determined by evaluation of the dependence of specific areas of run-off on the size of the river basin in historical flooding events in CZ—represents the position of the proposed test criterion. Hydrological units (watersheds) exceeding the criterion limit (points above the red line) are localities where there is exceptionally high justification for the establishment of permanent, structural measures. The precisely defined criterion means that socio-economic conditions in particular are considered, i.e., the problem moves to a level of political decision making. Figures 6 and 7 correspond to Figures 8 and 9.

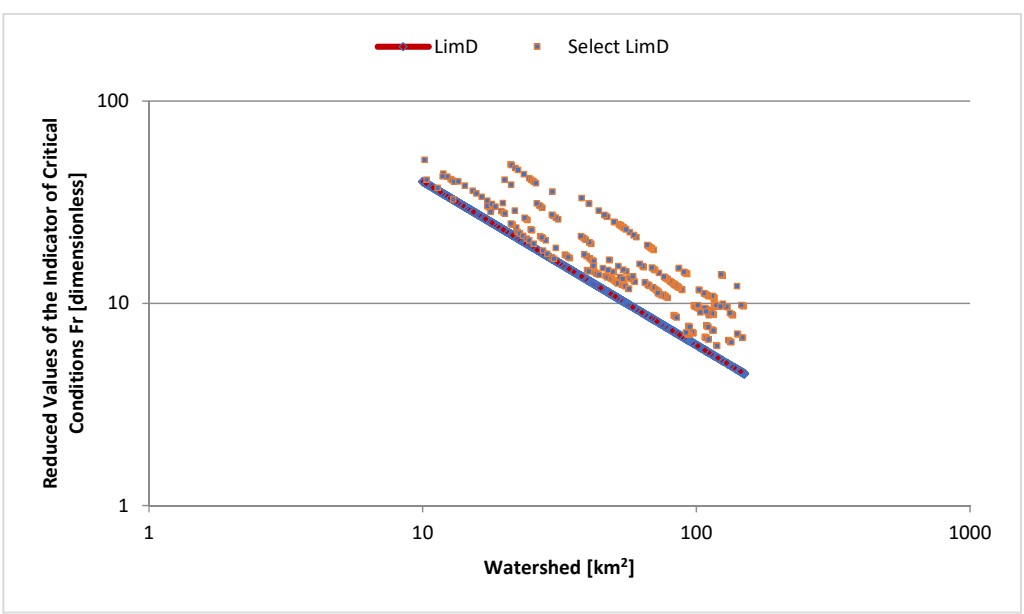

**Figure 7.** Selection of localities according to the testing criterion (5) within the group of watersheds 201–202–203.

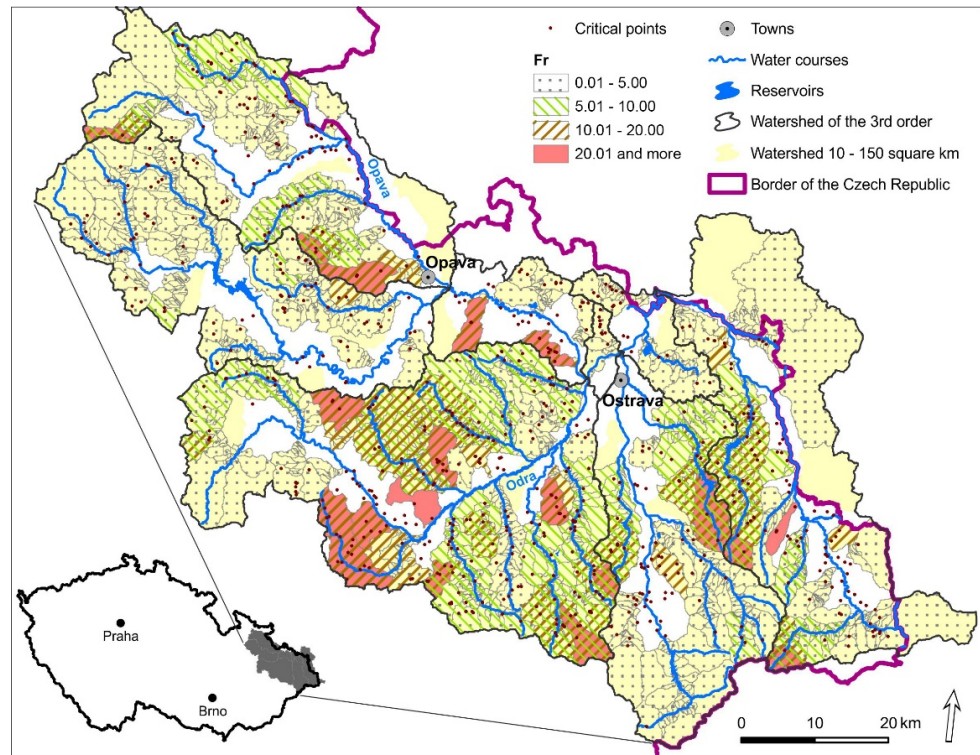

**Figure 8.** The group of watersheds 201–202–203 with categories of reduced value of weighted average of the indicator of critical conditions $F_r$.

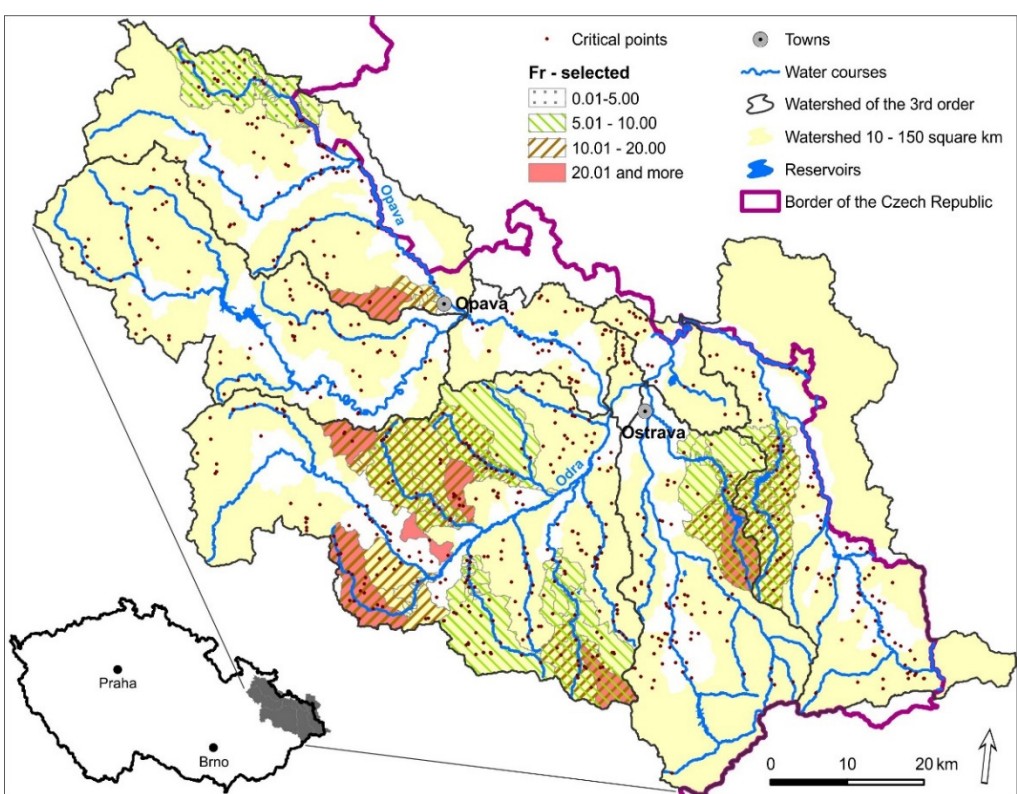

**Figure 9.** The selection of localities (group of watersheds 201–202–203) according to testing criterion (5).

Figures 7 and 9 represent the final selection of hydrological units according to the criterion (5). The selection criterion (red line) in a logarithmic scale, which is shown in Figure 7, can be expressed by Equation (5).

$$\ln(F_r) = a_1 \cdot \ln(A_w) + a_2 \tag{5}$$

where $F_r$ is the reduced value of the weighted average of the indicator of critical conditions (dimensionless variable), $A_w$ is the size of the watershed (km²), and $a_i$ (i = 1, 2) is the vector of straight-line parameter ($a_1 = -0.809$; $a_2 = 5.55$).

An example for the group of watersheds 201–202–203 is shown in Figure 8.

In Figure 9, the hydrological units that, by means of the reduced value of the weighted average of the indicator of critical conditions, fulfilled criterion (5) are shown. The differentiation in highlighting therefore expresses differences in $F_r$ values.

The evaluation was done throughout the Czech Republic. Calculations were based on the ratio of the sum of the parameters of the critical state weighted by the size of the river basin to the area of the highest hydrological unit (river basin). This means that reduced values of the weighted average of the indicator of critical conditions were calculated for all localities where CPs were generated. The overall initial results are shown in Table 4, where Figure 1 corresponds to the denomination of groups of watersheds.

Sections of watercourses and partial watersheds were included in the area of interest with the aim of creating an effective mitigation of floods and erosion events. Some examples obtained on the basis of primary results are illustrated in the following maps.

Figure 10 highlights selected hydrological units, which represent a recommended prioritization of enforcement and application of flash flooding mitigation measures.

**Table 4.** Overall results of selected variables for the Czech Republic (min. and max. are the lowest and the highest recorded values, respectively, of the respective variable in a group of watersheds).

| Groups of Watersheds | F-WA | | R | | $F_r$ | | S-Area (%) |
|---|---|---|---|---|---|---|---|
| | Min | Max | Min | Max | Min | Max | |
| 101–102–103 | 1.299 | 69.885 | 0.006 | 1.000 | 0.024 | 67.808 | 19.75 |
| 104–105 | 1.746 | 71.933 | 0.006 | 0.997 | 0.018 | 70.638 | 32.00 |
| 106–107 | 1.903 | 76.899 | 0.007 | 0.994 | 0.021 | 74.856 | 20.16 |
| 108 | 1.865 | 61.759 | 0.011 | 0.998 | 0.046 | 55.804 | 15.65 |
| 109 | 1.527 | 80.373 | 0.010 | 0.997 | 0.026 | 78.914 | 21.94 |
| 110–111 | 1.051 | 60.998 | 0.008 | 0.999 | 0.018 | 55.228 | 13.58 |
| 112 | 2.179 | 69.418 | 0.010 | 0.996 | 0.031 | 55.610 | 42.00 |
| 113a | 1.356 | 40.382 | 0.006 | 0.969 | 0.020 | 30.000 | 0.00 |
| 113b | 2.486 | 52.355 | 0.021 | 0.925 | 0.052 | 43.093 | 5.25 |
| 114 | 1.567 | 48.809 | 0.018 | 0.983 | 0.062 | 24.130 | 2.57 |
| 115 | 1.691 | 16.734 | 0.016 | 0.999 | 0.068 | 12.870 | 0.00 |
| 201–202–203 | 2.125 | 82.001 | 0.004 | 0.994 | 0.010 | 51.185 | 28.82 |
| 204 | 1.898 | 36.280 | 0.017 | 0.996 | 0.056 | 32.717 | 14.25 |
| 401–402 | 1.672 | 19.600 | 0.072 | 0.974 | 0.120 | 19.091 | 0.00 |
| 403–404 | – | – | – | – | – | – | – |
| 410–411 | 2.138 | 63.276 | 0.015 | 0.999 | 0.053 | 49.174 | 16.79 |
| 412 | 1.835 | 81.285 | 0.010 | 0.999 | 0.042 | 69.225 | 15.60 |
| 414 | 1.471 | 79.793 | 0.016 | 0.999 | 0.053 | 71.492 | 24.80 |
| 415 | 1.859 | 55.484 | 0.020 | 0.999 | 0.064 | 53.732 | 40.84 |
| 416 | 2.289 | 52.638 | 0.016 | 0.995 | 0.051 | 44.624 | 25.43 |
| 421 | 2.091 | 19.540 | 0.014 | 0.974 | 0.057 | 17.528 | 0.00 |

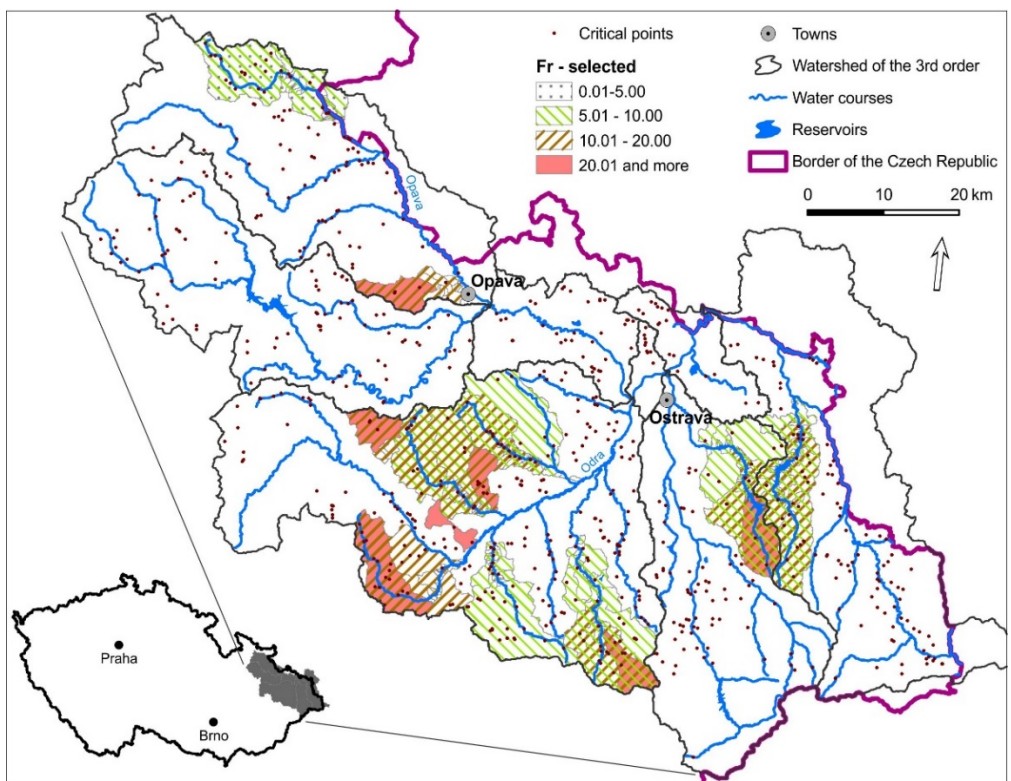

**Figure 10.** Selection of localities (group of watersheds 201–202–203) according to testing criterion (5) and sections of watercourses in areas of significant flood risk.

Hydrological units were determined on the basis of the aggregation of hazard data resulting from critical conditions contributing to areas of all identified CPs in model basins.

The described method is very effective and identifies in great detail the threat of concentrated runoff intrusion into a built-up area. It gives municipalities information on where the critical places are and why they are critical, e.g., having a low-capacity recipient or a culvert. Municipalities may ensure that no dangerous objects that could be transported to the built-up area during a concentrated outflow are stored in CP contributing areas. It also allows municipalities to influence the way these areas are managed by land users. Presented procedures were verified by professional, public, and state administration authorities; the results can be used for the Czech flood system, and river basin companies can incorporate them into flood risk management plans.

## 4. Conclusions

In the critical point method described in the study, data on concentrated surface run-off from torrential rain, throughout the Czech Republic, was used to identify and determine areas at risk (especially in urban zones). On the basis of the application of criteria described in this contribution, 9261 critical points (CPs) were identified. The total extent of catchment areas of selected CPs linked to built-up areas of municipalities in the Czech Republic is 18,112.2 km$^2$. The number of CPs is growing due to inappropriate urbanization, with new buildings being placed in the CP profiles. Therefore, a parameter-based aggregation was performed for further use, indicating the ratio of the sum of the critical condition parameters weighted by the size of the catchment areas to the highest hydrological unit area (watershed).

Using a weighted indicator of critical conditions applied to a watershed of the fourth order, a regionalization of the level of risk to the territory by flooding from torrential rain was carried out, again for whole country. Almost 35% of the state (approx. 27,500 km$^2$) lies outside the area of significant risk of this kind, and a further 40% of the territory has only a low level of risk from floods caused by torrential rains. Less than a quarter of the territory of the Czech Republic falls within the categories of medium (18.3%) and high (5.7%) level of risk.

It is obvious that protection against the negative effects of flash flooding is very difficult. Given the extent of the risk areas, it is not realistic to protect all critical locations equally. However, effective prevention could mean securing the buildings and infrastructure at risk, if not against flooding then at least from the dynamic effects of running water. The key task of preventing danger to human life and health might be aided also by updated flood plans with included CP localization information. The outputs of the study have already provided a nationwide basis for decision making by national and local authorities, as well as for planners of soil and water conservation measures.

Web-presentation of the critical points in the Czech Republic is available at http://webmap.dppcr.cz/dpp_cr/povis.dll?MAP=rizika_prival (accessed on 9 February 2022) POVIS flood information system, The Ministry of the Environment). The first version for Slovakia is available at https://lnk.sk/xwby (accessed on 9 February 2022) (URANOS, Faculty of Horticulture and Landscape Engineering, Slovak University of Agriculture in Nitra). Geoportals provide maps, from which the location of CPs and their contributing areas can be seen.

**Author Contributions:** Conceptualization, K.D., M.D. and B.Š.; methodology, K.D. and M.D.; formal analysis, K.D., M.D., B.Š., V.S., P.Š. and Z.M; investigation, K.D., M.D. and B.Š.; writing—original draft preparation, K.D., M.D., B.Š., V.S., P.Š. and Z.M.; writing—review and editing, K.D., M.D. and B.Š.; project administration, M.D., V.S. and P.Š. All authors have read and agreed to the published version of the manuscript.

**Funding:** This research was funded by the project No QK21010328: Potential for the development of small water reservoirs in the landscape as adaptation measures to eliminate hydrometeorological extremes, supported by the Ministry of Agriculture; the project No ITMS2014+ 313011W580: Scientific support of climate change adaptation in agriculture and mitigation of soil degradation, supported by the Integrated Infrastructure Operational Programme funded by the ERDF; and the project No SS02030018: Center for Landscape and Biodiversity, supported by the Technology Agency of the Czech Republic.

**Institutional Review Board Statement:** Not applicable.

**Informed Consent Statement:** Not applicable.

**Data Availability Statement:** Not applicable.

**Conflicts of Interest:** The authors declare no conflict of interest.

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
