# Peer review of "Mitigation of Flood Risks with the Aid of the Critical Points Method"

_agronomy, doi:10.3390/agronomy12061300_

Round 1

Reviewer 1 Report

The article is devoted to the important topic of assessing the risk of flooding of urban areas, which is especially important in the context of global climate change, which leads, among other things, to changes in the amount and intensity of precipitation. The article presents, according to the authors, the original methodology for assessing the risk of flooding of territories. A flood risk map of the Czech Republic has been constructed. At the same time, the title of the article (Mitigation of Soil Erosion, Sediment Transport and Flood 2 Risks with the Aid of Critical Points Method) does not reflect the essence of the work, since the work does not evaluate Soil Erosion and Sediment Transport. In my opinion, there is no need to write about Soil Erosion and Sediment Transport either in the Abstract or further in the text. I'm not a native English speaker, but I don't think the article is written in very good English. This leads to the appearance of sentences in the text that are very difficult to understand. The Introduction section needs to be redrafted to include method development as a goal, as well as an overview of the world's existing methodologies for assessing flood risk from extreme storm events. The Study area section needs to be expanded (details below). No section, original data. There are many questions to the methodology section that are not answered in the article (details below). The "Results and Disussion" section also needs to be redone to strengthen the discussion and comparison with the results of other studies. The "Conclusion" section also needs to be redone, adding the results of the work and removing the summary of the study.

43-46 “It is expected that, in CZ, extreme climatic phenomena will become more frequent,

with unusually dry periods, due to increased air temperature and a lack of precipitation,

as  well  as  intense  downpours  affecting  the  landscape  and  causing  environmental”

Based on what data do you make this statement? References are needed to climate change studies in the CZ in particular and the European Union in general.

46 «different approach» ??  May be “various approaches”. It seems to me that this is an important section, so you need to write what methods for assessing the risk of flooding currently exist, you need to write here in more detail.

51 “(2009, 2010 and 2013).”- What are these numbers here? Years? What do they mean? Why is the dot here?

55-56 I believe that here it is necessary to geographically expand the survey of flood risk assessment at least to the limits of the European Union, and not be limited only to the Czech Republic and Slovakia. And it is optimal to give an overview of such estimates for different continents over the past 5-10 years. I would also like to see here an assessment of the effectiveness of flood forecast data.

58-59 “Flash flooding due to torrential rainfall occurs mainly in small watersheds and is quite infrequent in CZ” аnd above you say the opposite.

60-61 Rephrase please

62-72 Here it is explained in more detail why it is important to assess the risk of flooding of territories, although this has already been mentioned above (32-42) in other words.In our opinion, these two parts should be combined.

70-71”Infrastructural measures [24], in the form of objects to reduce flooding, are mainly considered” By whom and where? Unclear.

81 «receptors» Not clear for me? What does it mean?

94-97 I do not think that it is worth giving the basics of hydrology here. Please explain why this is important?

108-110 very confusingly written in my opinion, difficult to understand. Please rephrase.

In addition, I believe that in this section there is very little information about the study area.Is the entire territory of the Czech Republic subjected to research or some part of it?There is no characteristic of the natural conditions that determine the formation of floods during precipitation: the physical properties of soils, the amount of humus, the average annual amount of precipitation, the intensity of precipitation capable of forming surface runoff, relief, land use (the percentage of plowing, crop rotations, etc.)

116 it is not clear why 0.3 and 10 km2 are taken as threshold values. Lines 117-120 do not provide sufficient explanation.

132 How the 3.5% slope threshold figure was calculated

133 there is no explanation for the threshold figure for the plowing area.

139 a1, a2, a3 – how were the model parameters obtained, why such figures?

Why does the equation not take into account the water saturation parameters of the soil cover?

141 “saturation” - the saturation of which?

145 Do I understand correctly that if F>=1.85 then we can call the point critical? Where did the threshold figure of 1.85 come from?

147 Hm,rhow was it calculated?

149 what kind of variables K1, K4 is not explained anywhere.

170 how the parameter a=0.536 was obtained

There is no chapter on the source data, so it is not at all clear where the source data and their spatial accuracy were taken from. Accordingly, it is not clear how the boundary of the watersheds, their average slope, etc. were determined.

156 As I understand it, this chapter is trying to transfer the results obtained for the watersheds of critical points to watersheds of a higher order?

168 What sample size was used to determine Equation 2 and also to find R2?

222-225 it is clear that there will be functional relationships between F and the catchment area, since they are included in one formula (1). The inverse relationship between F and the number of points CP is also understandable, since the smaller the catchment area, the greater their number and, accordingly, the greater the number of critical points.

229 Why is the threshold set to 15?

238-241 On what basis can this statement be made?

246-253 What values of Fr-corresponds to "high", "Middle", "Low" "Without any risk" the rate of risk and why?

255-257 The article states that the results of risk zoning have been tested in the three test catchments, but the results of this review are not included in the article.

260-266 How did you understand this, is it not clear? What methodology was used to assess impact? What soil and water protection measures do you mean?

315 «whole land.» not clear what is meant here?

323-328 It is very difficult to understand what is written here. Please rephrase.

Figure 10 shows sections of rivers with a potentially high risk of flooding (red), but it is not clear from the article how they were selected based on what criteria.

All figures  require a coordinate grid using longitude and latitude, the names of several settlements, or several names of major rivers.

334-335 Please rephrase.

336-345 Structurally, this is necessary in the introduction. And here, at the end of this section, give an analysis of how effective the approaches proposed in the world were in comparison with the methodology you propose.

336-337 «The approach presented in this study is an original one to the authors’ knowledge aiming at strengthening the base for flooding mitigation management». If the article proposes an original approach, then one of the tasks should be to announce the development of a methodology and present it in the article in more detail.

347-356 These are not conclusions, this is a summary of the work done, it is not necessary here.

357-359 These results were not obtained within the framework of this study. Are they needed here? "Critical points", "problem points" are they the same thing? Please be consistent.

360-364 The “Conclusion” section contains links to two geoportals showing the results of work to identify critical points within the Czech Republic and Slovakia, however, it is impossible for the reviewer to evaluate the full effectiveness of these resources, since the resources are not translated into English.

371 «their incidence» what exactly is meant?

372-382 All this is also not the results obtained in the article.

The article received a number of results, which for some reason were not included in the "Conclusion" section. Do the authors consider them not significant (for example, the distribution of the territory of the Czech Republic according to the risk of flooding)?

Author Response

Response to Reviewer 1 Comments

Authors thank the reviewer for valuable comments and tried to incorporate them as much as possible in the given time. All changes are highlighted in the text.

Title of article

Mitigation of Soil Erosion, Sediment Transport and Flood Risks with the Aid of Critical Points Method does not reflect the essence of the work, since the work does not evaluate Soil Erosion and Sediment Transport.

Title changed: Mitigation of Flood Risks with the Aid of Critical Points Method

Point 1 

43-46 “It is expected that, in CZ, extreme climatic phenomena will become more frequent, with unusually dry periods, due to increased air temperature and a lack of precipitation, as  well  as  intense  downpours  affecting  the  landscape  and  causing  environmental”

Based on what data do you make this statement? References are needed to climate change studies in the CZ in particular and the European Union in general.

Response 1

Added to the text.

Point 2 

46 «different approach» ??  May be “various approaches”. It seems to me that this is an important section, so you need to write what methods for assessing the risk of flooding currently exist, you need to write here in more detail.

Response 2

Added to the text.

Point 3 

51 “(2009, 2010 and 2013).”- What are these numbers here? Years? What do they mean? Why is the dot here?

Response 3

Reworded:

It was during the flood in 2009 that more than 100 localities were identified in the Jičínka and Luha river basins, where built-up areas were affected by surface runoff [5-7].

Point 4 

55-56 I believe that here it is necessary to geographically expand the survey of flood risk assessment at least to the limits of the European Union, and not be limited only to the Czech Republic and Slovakia. And it is optimal to give an overview of such estimates for different continents over the past 5-10 years. I would also like to see here an assessment of the effectiveness of flood forecast data.

Response 4

Added and thank you for the inspiration in this comment for further research.

Point 5 

58-59 “Flash flooding due to torrential rainfall occurs mainly in small watersheds and is quite infrequent in CZ” аnd above you say the opposite.

Response 5

Thank you, the typo has been corrected.

quite infrequent replaced by quite frequent

Point 6 

60-61 Rephrase please

Response 6

Reworded.

Point 7 

62-72 Here it is explained in more detail why it is important to assess the risk of flooding of territories, although this has already been mentioned above (32-42) in other words.In our opinion, these two parts should be combined.

Response 7

Combined and placed at the beginning of the introduction.

Point 8 

70-71”Infrastructural measures [24], in the form of objects to reduce flooding, are mainly considered” By whom and where? Unclear.

Response 8

Added what is meant by Infrastructural measures in terms of water management and anti-erosion measures.

Point 9 

81 «receptors» Not clear for me? What does it mean?

Response 9

Receptor replaced by “characteristics of contributing area”.

Point 10 

94-97 I do not think that it is worth giving the basics of hydrology here. Please explain why this is important?

Response 10

The mentioned text was included in the article to explain the established method of river basin division in the Czech Republic. 2nd, 3rd and 4th order river basin identifications are used in figures and tables in the text. The calculations themselves were organized by 2nd order groups.

Point 11 

108-110 very confusingly written in my opinion, difficult to understand. Please rephrase.

In addition, I believe that in this section there is very little information about the study area.Is the entire territory of the Czech Republic subjected to research or some part of it?There is no characteristic of the natural conditions that determine the formation of floods during precipitation: the physical properties of soils, the amount of humus, the average annual amount of precipitation, the intensity of precipitation capable of forming surface runoff, relief, land use (the percentage of plowing, crop rotations, etc.)

Response 11

A more detailed explanation has been added to the text.

Point 12

116 it is not clear why 0.3 and 10 km2 are taken as threshold values. Lines 117-120 do not provide sufficient explanation.

132 How the 3.5% slope threshold figure was calculated

133 there is no explanation for the threshold figure for the plowing area.

Response 12

Explanation added to the text:

The threshold values (size of the contributing area, slope conditions and percentage of arable land) were determined on the basis of field surveys and evaluation of hundreds of specific manifestations of damage in the built-up areas at particular critical profiles. A maximum area of 10 km2 was set on the basis of the limitation of the Curve Number (CN) method used.

Point 13

139 a1, a2, a3 – how were the model parameters obtained, why such figures?

Why does the equation not take into account the water saturation parameters of the soil cover?

Response 13

The parameters are part of a nonlinear regression relation (see equation 1). The parameters were obtained on the basis of an analysis of a set of data from hundreds of evaluated basins of critical points in the Czech Republic.

The soil moisture prior to a runoff event (AMC) parameters were taken into account. Indicator F was calculated for AMC II Antecedent Moisture Condition (AMCII): CNII - Average condition, built-up area hazards may occur at AMC II and more significant hazards occur at AMC III when soils in the drainage basins are practically saturated from antecedent rainfalls.

Point 14

141 “saturation” - the saturation of which?

Response 14

medium level of saturation replaced by AMCII- average level of saturation of soil

Point 15

145 Do I understand correctly that if F>=1.85 then we can call the point critical? Where did the threshold figure of 1.85 come from?

Response 15

Explanation added to the text.

The value of F means one of the conditions for the selection of the CP basin. The numerical value of 1.85 resulted from the evaluation of an extensive set of CP basin characteristics from the pilot areas of the Luha, Jičínka and Husí potok watercourses that were affected by the floods in 2009. The principles of the critical points method were tested on these data.

Point 16

147 Hm,rhow was it calculated?

 Response 16

Hm, r is the value from the central national database available for the entire territory of the Czech Republic, from CHMI data. Link to the database added to the text https://www.chmi.cz

Point 17

149 what kind of variables K1, K4 is not explained anywhere.

Response 17

Thanks for the notification. Typing error. Fixed to C1-C4.

Point 18

170 how the parameter a=0.536 was obtained

Response 18

The LTS regression method was used to determine the parameters of equations (2), (3) and (4).

Added to the text.

Point 19

There is no chapter on the source data, so it is not at all clear where the source data and their spatial accuracy were taken from. Accordingly, it is not clear how the boundary of the watersheds, their average slope, etc. were determined.

Response 19

Added to the text, also in accordance with point 11.

Point 20

156 As I understand it, this chapter is trying to transfer the results obtained for the watersheds of critical points to watersheds of a higher order?

Response 20

Yes, the models presented by equations (2), (3) and (4) were investigated (assessed) depending on the given quantities, especially on the degree of urbanization and runoff characteristics of higher order hydrological river basins. The data of the Czech Hydrometeorological Institute (Qn values in the profiles of water measuring stations) were used for this, and files with hydrological data were prepared for statistical evaluations, which represented the catchment area of water measuring stations with an area of up to 150 km2.

Point 21

168 What sample size was used to determine Equation 2 and also to find R2?

Response 21

Added to the text.

The LTS regression (LTS) method was used to determine the parameters of equations (2), (3) and (4). These three basic robust models were used to estimate the weighted average of the critical conditions indicator for hydrological units with available Qn values.

Point 22

222-225 it is clear that there will be functional relationships between F and the catchment area, since they are included in one formula (1). The inverse relationship between F and the number of points CP is also understandable, since the smaller the catchment area, the greater their number and, accordingly, the greater the number of critical points.

229 Why is the threshold set to 15?

Response 22

Added to the text.

For example, in an area with an F indicator lower than 15, it is not necessary to propose technical measures, “Best management practices”, such as anti-erosion agrotechnology, grassed waterways or protective grassing / afforestation, are sufficient to reduce the peak flow. The threshold value was set on the basis of hydrological calculations performed on hundreds of CP contributing areas.

Point 23

238-241 On what basis can this statement be made? The analysis of soils and localities endangered by torrential rain, or more precisely by flooding caused by torrential rain, shows that the overall size of these endangered areas is relatively high and for the whole of the Czech Republic it approximates to thousands of square kilometers.

Response 23

Based on the evaluation of data available for the entire territory of the Czech Republic, i.e. relief characteristics (DMR), pedological data from the pedological database and precipitation (CHMI).

Point 24

246-253 What values of Fr-corresponds to "high", "Middle", "Low" "Without any risk" the rate of risk and why?

Response 24

It is an evaluation of all 4th order river basins in the Czech Republic in connection with and in relation to the degree of urbanization of the area. According to indicator F, on 35% of the territory there are no CP basins, another 40% has low F.

Added to the text: Almost 35 % of the state (approx. 27,500 km2) lies outside the area of significant risk of this kind and a further 40 % of the territory has only a low level of risk from floods caused by torrential rains. Less than a quarter of the territory of the Czech Republic falls within the category of medium (18.3 %) and high level of risk (5.7 %).

Point 25

255-257 The article states that the results of risk zoning have been tested in the three test catchments, but the results of this review are not included in the article.

Response 25

The results of the evaluation of these river basins after the floods in 2009 were the basis for the CP method and the results were reflected in the setting of the limits of the basic parameters. The model river basins represent areas that were heavily affected by torrential rains in 2009, areas with high material damage and also loss of life.

These are pilot watersheds of the Luha, Jičínka and Husí potok, which we also mention in the text.

Point 26

260-266 How did you understand this, is it not clear? What methodology was used to assess impact? What soil and water protection measures do you mean?

Response 26

We have in mind organizational measures (grassing and afforestation of endangered areas), anti-erosion agrotechnology, and technical measures (protective reservoirs, dams, ditches, etc.). If all measures are built in the catchments of CP hydrological units 10–150 km2, we have proved that changes in runoff characteristics will be clearly evident. This resulted from a survey of a set of river basin data with monitored hydrological characteristics of the CHMI.

Point 27

315 «whole land.» not clear what is meant here?

Response 27

«whole land» replaced by throughout the Czech Republic

Point 28

323-328 It is very difficult to understand what is written here. Please rephrase.

Response 28

Reworded.

Point 29

Figure 10 shows sections of rivers with a potentially high risk of flooding (red), but it is not clear from the article how they were selected based on what criteria.

Response 29

The image has been modified.

Sections of rivers with a potentially high risk of flooding (red) have been removed from the picture. They are not necessary for the actual article dealing with floods from torrential rainfall in CP profiles.

Point 30

All figures  require a coordinate grid using longitude and latitude, the names of several settlements, or several names of major rivers.

Response 30

FIG. 1 defines the geographical position of the Czech Republic in the European area for the needs of the article. The coordinate network is not a benefit for the average reader for locating the displayed area.

FIG. 2 This is an illustration diagram. The grid is not useful.

FIG. 3 Illustrates the division of the territory of the Czech Republic into individual regions. A grid is not necessary. The location of the Czech Republic in a broader context is shown in Fig. 1.

FIG. 8 - 10 The location of the pilot area within the Czech Republic was added, two large cities in the region are shown, and the two most important rivers are described.

Point 31

334-335 Please rephrase.

Response 31

Rephrased.

Hydrological units were determined on the basis of the aggregation of hazard data resulting from critical conditions contributing to areas of all identified CPs in model basins.

Point 32

336-345 Structurally, this is necessary in the introduction. And here, at the end of this section, give an analysis of how effective the approaches proposed in the world were in comparison with the methodology you propose.

336-337 «The approach presented in this study is an original one to the authors’ knowledge aiming at strengthening the base for flooding mitigation management». If the article proposes an original approach, then one of the tasks should be to announce the development of a methodology and present it in the article in more detail.

Response 32

Modified in accordance with the comment. Text 336-345 moved to Introduction and reformulated ...

Point 33

347-356 These are not conclusions, this is a summary of the work done, it is not necessary here.

Response 33

Deleted.

Point 34

357-359 These results were not obtained within the framework of this study. Are they needed here? "Critical points", "problem points" are they the same thing? Please be consistent.

Response 34

Deleted.

Point 35

360-364 The “Conclusion” section contains links to two geoportals showing the results of work to identify critical points within the Czech Republic and Slovakia, however, it is impossible for the reviewer to evaluate the full effectiveness of these resources, since the resources are not translated into English.

Response 35

We have made the mentioned notes on web portals in order to schematically approach the issue graphically as well. From the Geoportal it is possible, also in national languages, to study map materials, from which the location of CP and their contributing areas is evident. A paper is already being prepared, which will focus on the presentation of web portals, where the relevant parts will also be translated into English. Thank you for your interest in such a topic.

Point 36

371 «their incidence» what exactly is meant?

Response 36

Rephrased:

Number of CPs is growing due to inappropriate urbanization, when new buildings are placed in the CPs profiles.

Point 37

372-382 All this is also not the results obtained in the article.

The article received a number of results, which for some reason were not included in the "Conclusion" section. Do the authors consider them not significant (for example, the distribution of the territory of the Czech Republic according to the risk of flooding)?

Response 37

Conclusions have been modified.

Reviewer 2 Report

Abstract need to be detailed particularly the results

map and limits of classes have to be corrected

discussion more developped 

Author Response

Response to Reviewer 2 Comments

Authors thank the reviewer for valuable comments and tried to incorporate them as much as possible in the given time. All changes are highlighted in the text.

Point 1

Abstract need to be detailed particularly the results

Response 1

Modified also in accordance with the other reviewer's comments.

Point 2

map and limits of classes have to be corrected

Response 2

Modified also in accordance with the other reviewer's comments.

Point 3

discussion more developped 

Response 2

Modified also in accordance with the other reviewer's comments.
